# EEG Network Analysis of Depressive Emotion Interference Spatial Cognition Based on a Simulated Robotic Arm Docking Task

**DOI:** 10.3390/brainsci14010044

**Published:** 2023-12-31

**Authors:** Kai Yang, Yidong Hu, Ying Zeng, Li Tong, Yuanlong Gao, Changfu Pei, Zhongrui Li, Bin Yan

**Affiliations:** 1Henan Province Key Laboratory of Imaging and Intelligent Processing, PLA Strategy Support Force Information Engineering University, Zhengzhou 450001, China; ykfer09@163.com (K.Y.);; 2Key Laboratory for Neuroinformation of Ministry of Education, University of Electronic Science and Technology of China, Chengdu 611730, China

**Keywords:** EEG, brain network, depressive emotion, spatial cognition

## Abstract

Depressive emotion (DE) refers to clinically relevant depressive symptoms without meeting the diagnostic criteria for depression. Studies have demonstrated that DE can cause spatial cognition impairment. However, the brain network mechanisms underlying DE interference spatial cognition remain unclear. This study aimed to reveal the differences in brain network connections between DE and healthy control (HC) groups during resting state and a spatial cognition task. The longer operation time of the DE group during spatial cognition task indicated DE interference spatial cognition. In the resting state stage, the DE group had weaker network connections in theta and alpha bands than the HC group had. Specifically, the electrodes in parietal regions were hubs of the differential networks, which are related to spatial attention. Moreover, in docking task stages, the left frontoparietal network connections in delta, beta, and gamma bands were stronger in the DE group than those of the HC group. The enhanced left frontoparietal connections in the DE group may be related to brain resource reorganization to compensate for spatial cognition decline and ensure the completion of spatial cognition tasks. Thus, these findings might provide new insights into the neural mechanisms of depressive emotion interference spatial cognition.

## 1. Introduction

Depressive emotion (DE), which is sometimes also called subthreshold depression, refers to clinically relevant depressive symptoms without meeting the diagnostic criteria for depression [1,2]. Studies have confirmed that individuals with depression show cognitive impairment, such as negative biases, slow response, low work efficiency, lack of concentration, and memory ability decline [3,4]. In recent years, more and more studies have paid attention to DE and found that individuals with DE also show cognitive impairment [5]. Liang et al. summarized the current research on the impact of subthreshold depression on autobiographical memory. They considered subthreshold depression as a typical negative emotion and found that most studies reported consistent results that subthreshold depression individuals have a negative bias in autobiographical memory [6]. Caterina et al. pointed out that depressive emotion can cause memory ability declines [7]. Yepez et al. assessed the executive function of older adults with depressive emotion via the Wisconsin Card Sorting Task (WCST), and individuals with depressive emotion achieved lower correct scores than healthy individuals [8]. Chen et al. investigated the cognitive control of individuals with depressive emotions in an emotional processing task based on event-related potential (ERP), and they found that cognitive control is significantly damaged in individuals with depressive emotions [9]. Li et al. reviewed recent research on depressive emotion and found that individuals with depressive emotions have abnormalities in cognitive activities, such as emotional processing, reward processing, and thinking activities [10]. As shown in Table 1, we can see that DE can also cause abnormal cognitive activities, so more attention should be paid to the cognitive processes of individuals with DE, due to its higher prevalence rate in the population.

Spatial cognition is a crucial cognitive ability that allows us to discern direction and perceive spatial distance relationships concerning others and objects [11,12,13]. In daily life, tasks, such as walking, driving cars, and pilots flying airplanes, all require normal spatial cognition [14,15]. Chen et al. found that patients with first-episode depression exhibit spatial cognition impairment. And, compared with the normal group, the electroencephalography (EEG) signal amplitudes of partial, occipital lobes increased significantly in the depression group [16]. They only focused on single brain regions, and the subjects were patients with depression. However, to the best of our knowledge, there have been few studies on the spatial cognition of individuals with DE. Moreover, spatial cognition ability is mediated by specific network connections and brain regions, such as the parietal lobe [17,18] and the frontoparietal network [19,20]. When DE individuals perform spatial cognition tasks, the characteristics of network connections remain unclear. This information can be conducive to understanding how DE affects spatial cognition and also can help diagnose DE. Therefore, it is necessary to analyze network patterns during the process of the spatial cognition tasks.

Network connections refer to the connections between brain regions that share function properties. In the past few years, studies have reported abnormal alterations in resting and task state network connections of DE and depression subjects. Moreover, many studies reported abnormal network connections and small-word properties in different frequency bands. For resting state networks, most differences in networks between the DE and HC groups were reported in theta and alpha bands. Hasanzadeh et al. found lower alpha connections between parieto-occipital regions of the depression group, and they also reported higher global efficiency in the depression group [21]. In contrast, Del’Acqua et al. reported on individuals with depression displaying increased theta and alpha connectivity between right frontal and central areas and right temporal and left occipital areas [22]. Hwang et al. also found increased default mode networks and control networks in subthreshold depression subjects [23,24]. For task state networks, the frontoparietal networks play an important role in spatial cognition tasks and gain increasing interest. In a visual word-learning game, frontoparietal networks facilitate word learning by directing the spatial attention of learners [25]. Douglas et al. found that decreased connectivity between the frontoparietal network and the rest of the brain is related to increased depression symptoms in undiagnosed individuals [26]. Moreover, Yeung et al. found that impaired cognitive control function in elderly individuals with depressive symptoms was associated with reduced activation of bilateral frontoparietal networks [27]. At the same time, researchers found that the enhancement in the frontoparietal network can compensate for the cognitive impairment of the depression group. Lee et al. found that when completing cognitive tasks of the same difficulty, the connectivity between the frontal and parietal lobes in the DE group was enhanced [28]. They believe that this is a brain restructuring to recruit more resources to perform executive functions, which can compensate for cognitive deficits in the brain. By analyzing and comparing the entropy of brain signals between the depression and healthy groups, Lin et al. found that there is a higher entropy in the left frontoparietal network of the depression group, which they believe reflects possible neural compensation during the treatment of depression [29]. In summary, DE can cause abnormal alterations in both resting and spatial cognition-related task networks. However, most studies only analyzed resting state or task state network connections of individuals with DE, and more insights can be gained into the networks of both resting and task states to analyze the network patterns of DE impacting spatial cognition.

**Table 1 brainsci-14-00044-t001:** Summary of studies about DE interference cognition.

Articles	Scales	Neural Signals	Resting State	Cognition Task	Brain Network	Frequency Band
[30]	DSM-IV	DTI	√	/	/	/
[31]	GDS-15	PET	/	executive function, memory	/	/
[32]	DSM-IV, CES-D	/	/	interview	/	/
[33]	BDI-II, OASIS, K10	EEG	/	Emotional processing	Coherence	√
[24]	CES-D, HAMD	MRI	√	/	general linear model	/
[23]	CES-D, HAMD	MRI	√	/	Seed-based	/
[7]	DSM-IV, HAM-D, FAST	/	/	verbal memory	/	/
[8]	MADRS	/	/	Wisconsin Card Sorting Task	/	/
[9]	BDI-II, SDS	EEG	/	Emotional processing	/	/
ours	CES-D, BDI-II	EEG	√	Docking task	PLV	√

The “√” indicates that the article contains this item, while the “/” indicates that the article does not include this item.

To explore the network patterns of DE impacting spatial cognition, this study first designed a simulated robotic arm docking experiment [34], which is close to the natural scene. Then, the resting and task state EEG were recorded from the DE and HC groups, and EEG networks and small-word properties of the network were calculated in different frequency bands. Finally, the differences in EEG networks and small-word properties of the network between DE and HC groups were compared and analyzed. The highlights of this study can be summarized as follows: (1) the brain network mechanism of DE impacting spatial cognition was explored in the task and resting states; (2) we designed a simulated robotic arm docking experiment close to a natural scene; (3) provided new insights for spatial cognition neural activity research on individuals with DE. The remainder of this paper is organized as follows: Section 2 describes the experimental design, data pre-processing, network construction, and analysis methods; Section 3 presents the results; Section 4 and Section 5 present our discussion and conclusions.

## 2. Materials and Methods

### 2.1. Subjects

To recruit subjects with DE, 76 undergraduate and graduate students aged 18–25 years from native universities were screened by advertisements. All subjects received an interview and surveys using the Center for Epidemiological Studies Depression Scale (CES-D, Chinese version) [35] and Beck Depression Inventory-II (BDI-II) [36]. The interview and survey results were carried out and assessed by professional psychology researchers. For subjects with depressive emotion, the inclusion criteria [2] were (1) a CES-D scale score greater than 16; (2) a BDI-II score no more than 13; (3) no history of being diagnosed with depression. Exclusion criteria [2,37] were (1) any suicidal tendencies; and (2) female subjects currently menstruating. For HC subjects, the inclusion criteria were (1) a CES-D scale score of less than 16; (2) a BDI-II score of no more than 13; (3) no history of being diagnosed with depression; (4) female subjects not currently menstruating.

Then, 15 subjects (6 females, mean age 22.5 ± 1.6) were screened for the depressive emotion (DE) group, and the other 15 subjects (7 females, mean age 22.3 ± 1.8) were selected as the HC group. The averaged CES-D scores and BDI-II scores of the DE group were 24.8 ± 4.2 and 10.9 ± 2.6, respectively. The averaged CES-D scores and BDI-II scores of the HC group were 14.6 ± 3.4 and 8.2 ± 2.1, respectively. The CES-D scores of DE group were significantly greater than the CES-D scores of the HC group (*p* < 0.05), and the BDI-II scores of the DE group were larger than the BDI-II scores of the HC group but did not meet the significance threshold (*p* > 0.05).

Due to the COVID-19 epidemic, all the subjects had been staying at school for more than 60 days. While at school, the activity scope and lifestyle of all the subjects were similar. The subjects mostly stayed in the dormitory every day. Four subjects were living in each dormitory, and the diet was the same every day. All subjects were right-handed, had normal vision or corrected normal vision, and normal intelligence. The experiment was approved by the Institution Research Ethics Board of the University of Electronic Science and Technology of China. The subjects were informed of the purpose and content of the experiment beforehand and signed an informed consent form according to the Declaration of Helsinki. After the experiment, each subject received a certain amount of experimental allowance.

### 2.2. Experimental Design

Here, this study aims to investigate the effect of depressive emotion on spatial cognition. The classical cognitive task paradigm, such as the Stroop paradigm [38], n-back paradigm [39], and emotional face processing paradigm [40], has the advantage of simple design, but these experimental paradigms not only fail to mobilize participants’ spatial cognition ability but are also far from natural scenes in our work or life. Robotic arm operation is a vital ability for workers in modern factories and even for astronauts. So, this study designed the simulated robotic arm docking task as an ecological paradigm for the research on spatial cognition ability. During the experiment, subjects need to mobilize their spatial cognition ability to observe and plan the spatial position of the robotic arm and operate it to make appropriate adjustments [41], which is in line with our experimental purpose.

The simulated robotic arm docking task experiment was designed on a virtual robot experimentation platform (V-REP) [42,43]. The experiment was conducted on an HP Z-book computer with Intel Core i9-9880H 2.3 GHz CPU (HP (Chongqing) Co., Ltd., Chongqing, China), 64 GB memory, and Windows 10 professional version operation system. The purpose of this experiment is for the subject to control the simulated robotic arm through obstacles by a mouse to complete the docking with the target point. During the experiment, subjects first need to observe the spatial position of the robotic arm and then adjust the spatial position of the robotic arm to make it close to the docking point. Thirdly, they need to fine-adjust the angle and height of the robotic arm and finally operate the robotic arm through obstacles to complete the docking task.

As shown in Figure 1, the experiment consists of three parts: preparation, resting state EEG recording, and formal experiment. The preparation contains the subject screen via depression scales and practical experiments. The practice experiment enables the subjects to master the contents of the simulated robotic arm docking task. The tasks in the practice experiment are similar to but different from the tasks in the formal experiment. The formal experiment was conducted within one week after the practice experiment. There are two types of obstacles in formal experiments: vertical bar obstacles and circular frame obstacles in front of the docking point; each type of obstacle has eight trials. To reduce the practice effect with the increase in the number of experiments, eight initial operating perspectives were set for the two obstacle situations, which were at eight positions of 30°, 60°, 90°, 120°, 150°, 180°, 210°, and 240° deviation from the middle vertical line of the horizontal interface, and the starting position was the same distance from the docking point. Therefore, there are 2×8 = 16 trials in formal experiments. All blocks have the same level of difficulty, and the order of the 16 trials is random. After each trial, there is a rest for at least 1 min. In the experiment, subjects can see the overall angle of view and three-angle views of the robotic arm. The subjects adjust the position of the robotic arm in the overall angle of view interface according to the other three views.

During the experiment, the operation details and operation time of the subjects while completing the docking task were recorded using screen recording software (EV screen recording 5.1.0). The time resolution of the screen recording video was 30 frames per second. The EEG signals were also recorded throughout the whole experiment.

### 2.3. EEG Signal Acquisition and Pre-Processing

The experiment was conducted in a quiet, electromagnetically shielded room with a temperature of around 23 °C and suitable lighting. In the experiment, the subjects sat in a comfortable chair with adjustable height and were about 60 cm away from the screen. The EEG acquisition system was the G.HIamp bio-signal acquisition system of Austria G.tec Company (Styria, Austria). The system could collect 63 channels of EEG signals. The sensors were positioned according to the 10–20 system. The 63rd channel was the right earlobe, which was a reference sensor, so there were 62 effective electrodes. The sampling rate of recorded signals was 600 Hz. Before the formal docking task experiment, resting state (RS) EEG signals were recorded while subjects sat quietly with their eyes closed for 2 min. Then, the subjects started the docking task experiment, and the EEG signals were recorded throughout the whole experiment.

During EEG signal acquisition, online filtering was performed with 0.1–100 Hz and 48–52 Hz band-pass filters. The offline preprocessing of EEG signals mainly included data segmentation (2 s interval), 0.1–60 Hz band-pass filtering, fast-ICA artifact removal [44], baseline correction, and average reference.

### 2.4. Operation Time Analysis

According to a previous study [45], the docking process is divided into four stages. The observation stage (O stage) is the period from pressing the start button to the time point of moving the entire robotic arm. The large-scale movement stage (LM stage) is the period from the time point of moving the entire robotic arm to the time point of fine adjusting the joint angle of the robotic arm. The fine operation stage (FO stage) is from the time point of fine adjusting the joint angle of the robotic arm to the time point when the subject moves the entire robotic arm again. The docking stage (D stage) is from the time point when the subject moves the entire robotic arm again to the moment when the docking task is completed.

The operation time of each operation stage was obtained through offline analysis of video recording. For each subject, the operation time of the same operation stage was averaged. For each group, the operation time of the same operation stage was averaged from all subjects of the same group. To investigate group differences, the averaged operation time in each stage and total time of the two groups (DE group and HC group) were compared by the Wilcoxon rank sum test.

### 2.5. EEG Network Analysis

Many previous studies have reported the differences between healthy and abnormal groups in frequency domain [46,47]. To explore the differences in network patterns between DE and HC groups in frequency bands, the preprocessed EEG signals were firstly filtered into delta (1–4 Hz), theta (4–8 Hz), alpha (8–13 Hz), beta (13–30 Hz), and gamma (30–50 Hz) bands using a Chebyshev filter [48,49,50]. The band-pass filtering process involves first passing through a low-pass filter and then through a high-pass filter.

Then, the EEG electrodes were considered the nodes of the network. The phase locking value (PLV), which reflects the synchronization of the EEG signals from every two electrodes, was used to determine the edges of the network. In each frequency band, there was one 62*62 matrix for each subject. For both the DE group and the HC group, there were 15 62*62 matrices. The Wilcoxon rank sum test was conducted on the matrices of two groups. The single matrix elements of the two groups were compared by Wilcoxon rank sum test with a sample size 15. The Wilcoxon rank sum tests were conducted 62*62 times, and we obtained a new 62*62 matrix. The elements of the new 62*62 matrix were *p*-values from the Wilcoxon rank sum tests. Then, the false-discovery rate (FDR) verification was utilized with threshold 0.05 to correct the results of the Wilcoxon rank sum test. Then, we set the value less than 0.05 in the matrix to 1 and the value greater than 0.05 to 0, so we transformed the matrix into a binary matrix. The element “1” means the network connection value of the DE group was greater than that of the HC group, and the element “0” means the network connection value of the HC group was greater than that of the DE group. The network connections can be shown in figures with different-colored lines according to the “0” and “1” in the binary matrix.

According to graph theory, an EEG network can be characterized by small-world property. So, the following small-world properties were calculated to depict the connectivity and information interaction efficiency of the network. The characteristic path length (CPL) and global efficiency (Ge) reflect the global information interaction of the network, and the clustering coefficient (CC) and local efficiency (Le) reflect the local connectivity and information interaction of the network. The node out-degree and in-degree were also calculated to evaluate the connectivity between nodes in the network. Because the PLV network is undirected, the out-degree and in-degree are the same and called node degree (ND) [21]. The calculation details can be found in our previous work [51].

Due to the data in this study not meeting the normal distribution conditions, the brain network connections and the property CC of the DE and HC groups were compared by the Wilcoxon rank sum test. For brain network connections, to avoid the deviation caused by multiple comparisons, false-discovery rate (FDR) [52] verification was utilized with threshold 0.05 to correct the results of the Wilcoxon rank sum test. In addition, the whole brain was divided into 10 sub-brains and the Spearman correlation coefficients were calculated between average property CC of each sub-brain and CES-D scores.

## 3. Results

### 3.1. Behavior Data Results

The average operation time in each stage of the DE and HC groups is shown in Table 2. According to the results, it can be seen that the operation time of the LM stage and FO stage and the total time of the DE group were significantly longer than those of the HC group (*p* < 0.05), while there was no significant difference between the two groups (*p* > 0.05) for the operation time in O stage and D stages.

### 3.2. Brain Network Connections with a Significant Difference

According to the behavioral results, there were significant differences in the operation time of the LM stage and FO stage between the two groups. So, for task state networks, this study focused on analyzing the differences in network connections between the DE and HC groups in the LM and FO stages.

The network connections with significant differences between the DE and HC groups in RS state, LM, and FO stages are shown in Figure 2. For the RS stage, network connections with significant differences between DE and HC groups were mainly in theta and alpha bands (Figure 2(B1,C1)). More specifically, a strong increase in the network connections of the HC group compared to the DE group in the theta band was found among frontal, central, and parietal regions. Concerning the alpha band, compared with the DE group, the HC group had stronger network connections between the right frontal and left parietal, and the hub nodes were located in the left parietal. For the task stage, network connections with significant differences between DE and HC groups were mainly in delta, beta, and gamma bands (Figure 2(A2,A3,D2,D3,E2,E3)). The network connections with significant differences in LM and FO stages show similar patterns. More specifically, compared with the HC group, the DE group has stronger connections between the left frontal and left parietal regions, and the hub nodes are located in the left frontal.

### 3.3. Difference in Small-Word Properties between Two Groups

According to the calculation methods, the CC, Le, Ge, and ND are positively correlated with each other, while the CPL is negatively related to the other four properties. The distributions of nodes with significant differences in CC, Le, Ge, CPL, and ND are almost the same. Therefore, this study only displays the results of CC.

Figure 3 shows the distribution of scalp positions where the CC has significant differences between the DE and HC groups in the RS state, LM, and FO stages. The results of the network property differences between the HC and DE groups are consistent with the results of the network connections. For the RS stage, the HC group has higher CC than the DE group, and the nodes with significant differences are mainly located in the left parietal region in the theta band (Figure 3(B1)). As for task stages, the DE group has higher CC than the HC group in some nodes, mainly in the delta, beta, and gamma bands. More specifically, for the LM stage, the DE group has higher CC in nodes, mainly located in the left frontal, left parietal, and occipital regions in the delta band (Figure 3(A2)), left frontal and central regions in the beta band (Figure 3(D2)), and left frontal, right parietal, occipital regions in the gamma band (Figure 3(E2)). For the FO stage, the DE group has higher CC in nodes, mainly located in the left frontal and left parietal regions in the delta band (Figure 3(A3)), left frontal region in the beta band (Figure 3(D3)), and left frontal and right parietal regions in the gamma band (Figure 3(E3)).

### 3.4. Relationship between Small-Word Properties and the CES-D Scores

The relationships between small-word properties and the CES-D scores were also analyzed. Firstly, the whole brain was divided into 10 non-overlapping sub-brain regions, namely the left frontal, right frontal, left central, right central, left temporal, right temporal, left parietal, right parietal, left occipital, and right occipital. The brain regions were divided, as shown in Figure 4. The property CC from the electrodes in each sub-brain region was averaged. Then, for each sub-brain region, the correlation coefficients between the averaged CC and the CES-D scores were calculated based on Spearman correlation.

Table 3 shows the correlation coefficients between the averaged property CC of sub-brain regions and the CES-D scores in each frequency band of the RS, LM, and FO stages. Generally, it can be seen that the property CC was negatively related to CES-D scores in the RS stage, and the property CC was positively related to CES-D scores in LM and FO stages.

More concretely, for the RS stage, the averaged CC from the LT region in the theta band, the averaged CC from the LC region in the alpha band, and the averaged CC from the LT region in the gamma band were negatively correlated with CES-D scores.

For the LM stage, the averaged CC from the LF region in delta, beta, and gamma bands was positively correlated with CES-D scores. The averaged CC from the LP region in the delta band and the averaged CC from the RP region in the gamma band were positively correlated with the CES-D scores. The averaged CC from bilateral occipital regions in delta and gamma bands was positively correlated with CES-D scores.

For the FO stage, the averaged CC from the LF region in delta, beta, and gamma bands was positively correlated with CES-D scores. The averaged CC from LT, LP, and LO regions in the delta band was positively correlated with CES-D scores.

On the whole, the results of correlation coefficients are consistent with those of different network connections and different properties. For example, the correlation coefficient of the averaged CC from the left frontal region was greater than that of the averaged CC from other brain regions in task stages. Moreover, it can be observed that the correlation coefficients of most left-brain regions are greater than those of right-brain regions, and the brain regions with larger correlation coefficients are mainly the left frontal, left parietal, and occipital regions.

## 4. Discussion

The current study explored the network connection alterations in resting and spatial cognition task states of individuals with DE. Our findings demonstrated that DE can cause impairments in spatial cognition, and the impairment is associated with weaker network connections in the RS stages of the DE group. Notably, enhanced left frontoparietal connections were observed in the DE group, which possibly reflected the compensation mechanism of DE individuals while completing spatial cognition tasks.

Depressive emotion impairs spatial cognition, resulting in longer operation time

The analysis of the behavioral data revealed a significant effect of DE on spatial cognition. The behavioral results showed that the operation time in the LM and FO stages of the DE group was significantly longer (*p* < 0.05) than those of the HC group. The docking tasks in the LM and FO stages mainly involve the spatial positioning movement of the entire robotic arm and the fine adjustment of the arm joint angle. These operations require the spatial cognition of subjects. Previous studies have pointed out that individuals with depression have a certain decline in their perception of surrounding things, and their spatial cognition ability is affected, resulting in poor cognitive performance [31,53]. In this experiment, compared with the HC group, the operation time in the LM and FO stages was longer for the DE group, which was consistent with the results in previous studies showing that DE can cause cognitive abilities to decline.

The weaker network connections in the RS stage of the DE group

In the RS stage, compared with the HC group, the network connections were weaker for the DE group, the network property CC was smaller for the DE group, and the averaged network property CC was negatively related to CES-D scores. Moreover, the differences were mainly in frontal and parietal regions in theta and alpha bands. The results were consistent with previous studies on resting-state network connections of depression [54]. The weaker connections in theta and alpha bands were most reported in previous studies [46], and the weaker connections might indicate increased depression symptoms [26]. In addition, the alterations in the frontal region were related to abnormal emotion processing of the depression group, such as negative emotion bias [55]. Therefore, the weaker connections among frontal regions in the RS stage might reflect depressive symptoms, negative emotion bias, and being down in the minds of the DE group [56,57]. The parietal region was related to spatial cognition [58,59]. Compared with the HC group, the DE group showed weaker connections and smaller network property CC in the parietal region, which might explain the spatial attention decline in the DE group.

The enhancement in frontoparietal connections in LM and FO stages of the DE group

In contrast to the results in the RS stage, enhanced left frontoparietal network connections were observed in the DE group. The frontoparietal network was a prominent network related to many cognition abilities, especially spatial cognition. Many previous studies have also observed enhanced frontoparietal connections in cognitive tasks of the cognitive impairment group, which is similar to our results. Smith et al. reported that compared with healthy individuals, the left hemisphere of individuals with depression was more active, mainly concentrated in the frontal and central parietal brain regions [60]. Ho et al. conducted a comparative analysis of the complexity of the network between individuals with depression and the healthy group and found that the complexity of the left frontoparietal network in depression patients was higher than that in the healthy group, which they believed was essential for executive control function [61]. Verga et al. reported that the increased frontoparietal networks enhanced visuospatial attention during interactive learning [25]. Sklar et al. also found greater frontoparietal networks in visual spatial search tasks in a first-episode schizophrenia group compared to the healthy group [62].

Moreover, the weaker connections in the RS stage and enhanced frontoparietal networks were observed in the LM and FO stages, and these contrast network patterns in RS and task stages were consistent with the brain resource reorganization and compensation mechanism reported in previous studies [63]. Lee et al. found that when completing cognitive tasks of the same difficulty, the connection between frontal and parietal lobes was enhanced in individuals with depression. They believed that this was the reorganization of the brain to recruit more resources to perform cognitive tasks, a mechanism that can compensate for the cognitive deficits of the brain [28]. In an emotional face-word Stroop task, researchers also found larger frontoparietal networks in the depression group compared to the HC group, and they thought this was caused by brain resource reorganization. Therefore, the different network patterns in RS and task state stages might indicate the brain resource reorganization and compensation mechanism of the DE group during spatial cognition tasks.

We also found that the differences between the DE and HC groups in task stages are mainly in delta, beta, and gamma bands. Many studies have reported that the increased connections in the beta and gamma frequency bands were related to cognitive activities [64]. Some studies also found increased delta band brain activities in cognitive processing. Particularly, when individuals engaged in math tasks and graph transformation, brain activity in the delta band increased [65,66]. The enhanced network connections in delta, beta, and gamma bands might reflect the unique frequency domain activity patterns of the DE group during docking tasks.

## 5. Conclusions

In this study, based on a simulated robotic arm docking experiment, the underlying differences in EEG networks between DE and HC groups were investigated in RS and task stages. The behavioral results indicated that DE interferes in spatial cognition, resulting in significantly longer operation time in LM and FO stages. In the RS stage, the DE group had weaker network connections and lower network efficiency in theta and alpha bands than the HC group had. The electrodes in parietal regions were hubs of the differential networks, which is related to spatial cognition and indicates the spatial cognition decline in the DE group. Furthermore, in the LM and FO stages, the DE group had stronger left frontoparietal network connections in delta, beta, and gamma bands than those of the HC group. The enhanced left frontoparietal connection in the DE group may be related to brain resource reorganization to compensate for spatial cognition decline and ensure the completion of docking tasks. Thus, these findings might provide new insights into neural mechanisms of depressive emotion interference spatial cognition.

## Figures and Tables

**Figure 1 brainsci-14-00044-f001:**
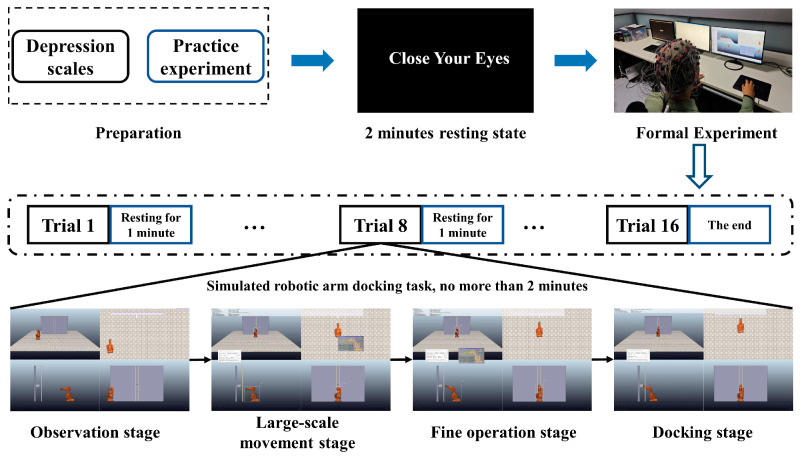
The experimental paradigm.

**Figure 2 brainsci-14-00044-f002:**
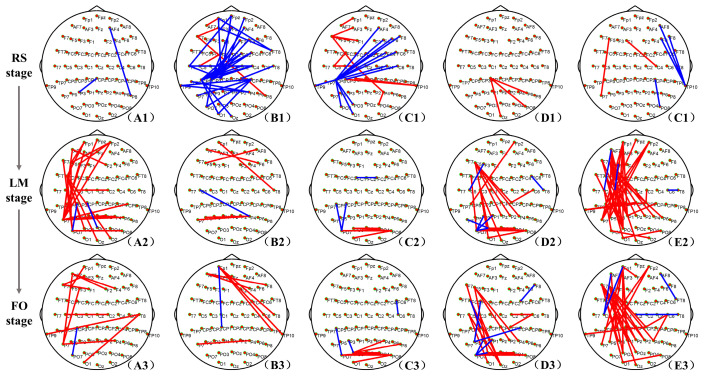
The network connections with significant differences between the DE and HC groups in the RS, LM, and FO stages. The red lines represent the significantly stronger network connections in the DE group compared to the HC group, and the blue lines represent the significantly stronger connections in the HC group compared to the DE group, *p* < 0.05. The (**A**–**E**) correspond to delta, theta, alpha, beta, and gamma bands, respectively.

**Figure 3 brainsci-14-00044-f003:**
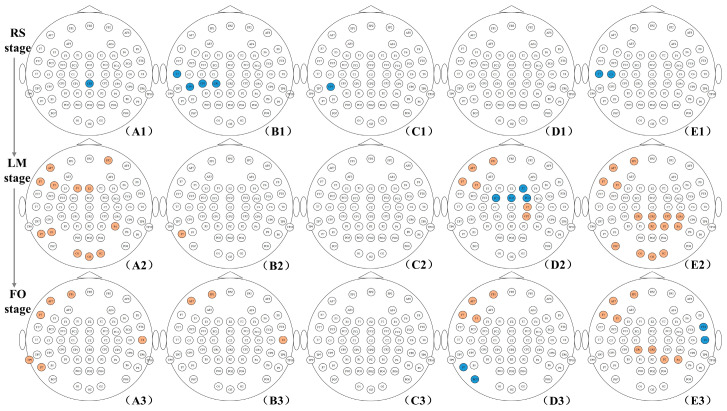
The distribution of scalp positions where the CC has significant differences between the DE and HC groups in the RS, LM, and FO stages (*p* < 0.05). The yellow electrodes indicate that the DE group has larger CC values compared to the HC group, while the blue ones show that the HC group has larger CC values. The (**A**–**E**) correspond to delta, theta, alpha, beta, and gamma bands, respectively.

**Figure 4 brainsci-14-00044-f004:**
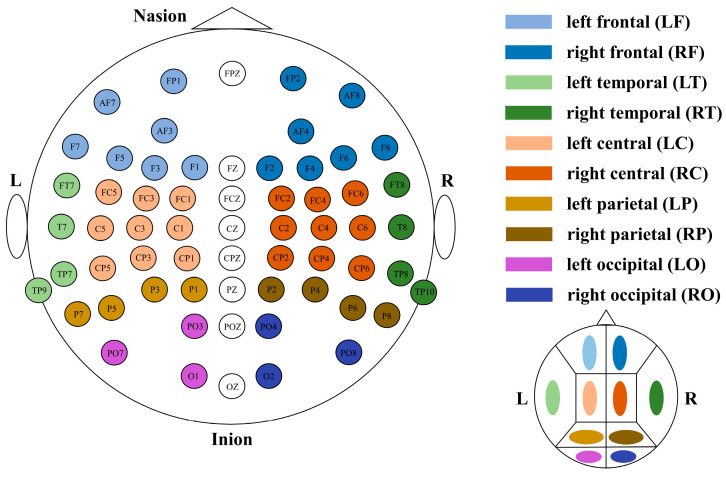
Segmentation of 10 sub-brain regions.

**Table 2 brainsci-14-00044-t002:** The average operation time in each stage of the two groups (s).

Groups	O Stage	LM Stage	FO Stage	D Stage	Total Time
HC	10.91 (2.6)	20.52 (3.2)	19.31 (1.9)	24.22 (3.8)	74.96 (5.6)
DE	11.53 (3.0)	25.68 (2.9)	26.47 (2.2)	25.75 (4.3)	89.43 (6.4)

The underline indicates that the operation time of the DE group is significantly longer than that of the HC group, *p* < 0.05.

**Table 3 brainsci-14-00044-t003:** The correlation coefficients between the averaged CC of sub-brain regions and the CES-D scores in each frequency band of the RS, LM, and FO stages.

	LF	RF	LC	RC	LT	RT	LP	RP	LO	RO
RS delta	0.13	0.06	0.12	0.10	0.02	−0.24	0.07	−0.19	0.04	0.17
RS theta	0.03	−0.07	−0.10	−0.20	**−0.31**	−0.15	−0.20	0.22	0.05	0.21
RS alpha	0.21	0.20	**−0.32**	0.14	0.06	0.16	−0.13	0.20	0.10	0.12
RS beta	0.02	−0.06	−0.03	−0.03	−0.11	−0.19	−0.06	0.07	0.01	0.14
RS gamma	0.08	0.01	0.00	−0.04	**−0.33**	−0.12	0.17	−0.13	0.07	−0.13
LM delta	**0.41**	0.23	0.22	0.22	0.16	0.27	**0.40**	0.19	**0.41**	**0.38**
LM theta	0.22	0.20	0.12	0.15	0.08	0.06	0.21	0.19	0.20	0.22
LM alpha	0.07	0.07	0.03	0.06	0.07	0.01	0.06	0.07	0.01	0.03
LM beta	**0.38**	−0.11	−0.06	0.05	−0.10	0.04	−0.12	−0.23	−0.20	−0.11
LM gamma	**0.35**	0.11	0.15	0.14	0.17	0.27	0.22	**0.33**	**0.36**	**0.33**
FO delta	**0.42**	0.24	0.27	0.15	**0.32**	0.17	**0.41**	0.21	**0.35**	0.19
FO theta	0.23	0.13	0.04	0.09	0.07	0.16	0.19	0.11	0.14	0.14
FO alpha	0.14	0.10	0.04	0.10	0.11	0.06	0.18	0.14	0.19	0.13
FO beta	**0.35**	−0.08	−0.12	0.02	−0.12	−0.05	−0.10	−0.03	−0.21	−0.11
FO gamma	**0.34**	−0.05	0.11	0.18	0.00	0.17	0.13	0.27	0.10	0.11

The bold values indicate a significant correlation (*p* < 0.05).

## Data Availability

The data presented in this study are available on request from the corresponding author. The data are not publicly available due to the project supporting this study is not yet completed, so the use of data needs to be approved by the project leader.

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
