# Peer review of "EEG Network Analysis of Depressive Emotion Interference Spatial Cognition Based on a Simulated Robotic Arm Docking Task"

_brainsci, 2023, doi:10.3390/brainsci14010044_

Round 1

Reviewer 1 Report (Previous Reviewer 3)

Comments and Suggestions for Authors

The authors present the article entitled “EEG Network Analysis of Depressive Emotion Interference Spatial Cognition Based on a Simulated Robotic Arm Docking  Task

This study analyzes the network patterns of DE impacting spatial cognition from resting and task state EEG networks: Designes a simulated robotic arm docking experiment close to a natural scene and Provides new insights for cognitive neural activity research on individuals with DE.

The article presents the following concerns:

  • Add hyperlinks to tables, figures, and references.

  • In this kind of text, apostrophes must be avoided. For example: learner's 

  • The abstract is too extended, please limit it to no more than 200 words

  • Avoid speaking in the first person singular, instead use passive voice

  • The abstract mentions the main contributions and quantitative and qualitative research results.

  • Remove the numbers from keywords.

  • Review as several acronyms are defined multiple times as healthy control (HC)

  • Lines 74 and 75 can be justified with the following references: Impact of eeg parameters detecting dementia diseases: a systematic review; A new approach for motor imagery classification based on sorted blind source separation, continuous wavelet transform, and convolutional neural network.

  • Spaces do not separate several types, such as references from the previous word. Please correct.

  • The controversy and the problems to be resolved regarding the advances and previous work need to be clarified. Please highlight how the main findings of this work contribute to the state of the art.

  • Why was it decided to use the Chebyshev filter to segment the signals and not, for example, a wavelet transform?

  • Add a comparative table of the results obtained with those of previous works.

  • Please update the references since less than 42% are greater than or equal to 2018.

  • A better analysis and description of Table 2 is required.

Comments on the Quality of English Language

The following misspellings should be checked:

  1. line 169: “ to first observe…” should be rewritten by “first to observe …”

  2. line 217: The word “whole” appears repeatedly in this text. Consider using a synonym in its place. 

  3. line 354: The word “importantly” is often overused. Consider using a more specific synonym like “notably” to improve the sharpness of your writing.

Author Response

Thank you for your kind opinion.  Please see the attachment

Reviewer 2 Report (Previous Reviewer 2)

Comments and Suggestions for Authors

The authors have addressed most of my previous comments.

Author Response

Thank you for your kind comments.

Reviewer 3 Report (Previous Reviewer 1)

Comments and Suggestions for Authors

Kai Yang et al. thoroughly revised their paper on EEG network analysis in humans during an operant cognitive task after peer review. The authors answered all my questions and provided a thorough response letter, but the problem with statistical analysis and with the overall reliability of their results remains.

Wilcoxon test. "We have conducted a Wilcoxon analysis on the CES-D and BDI scale scores of two groups." Why did the authors choose the test for dependent samples (Wilcoxon test) instead of the Mann-Whitney U-test tests for two independent samples?

The same problem in lines 225 -227: "time in each stage and total time of the two groups (DE group and HC group) were compared by paired Wilcoxon test." The samples DE and HC groups are independent, therefore, the paired Wilcoxon test is inappropriate here. Could the outcomes of ANOVA (parametric or non-parametric) for the factor "Group" (2 ways: DE/HC) and the factor "Stage" (4 ways: O/LM/FO/D) be reported?

In all, the results of all statistical tests should be reported in a clear, concise, and accurate way. For example, for the non-parametric Mann–Whitney test for independent samples, it is important to state: medians, the value of U, the sample sizes, the significance level, and the difference between the two samples.

The statistical analysis of "network connections" is unclear to me (lines 239-256).

As far as I understood, five characteristics were measured in 62*62 matrix of EEG signals. Please clearly explain what measures were reported as "commonly used network properties"? CPL) Ge) CC) Le) and ND). What measures are depicted in Figure 2?

Please revise the term "network properties", which sometimes implies association measures. For example, in Line 248 ("this paper calculated five commonly used network properties.."), also in Line 239 ("network properties that can describe the connectivity and efficiency" - properties could not describe), as well as in Line 251 ("the network properties of the DE and HC groups were compared by Wilcoxon test with threshold 0.05" - properties could not be compared by Wilcoxon test; 0.05 is the level of significance or type I error; please revise this sentence).

Author Response

Thank you for your kind opinion.Please see the attachment. 

Round 2

Reviewer 3 Report (Previous Reviewer 1)

Comments and Suggestions for Authors

The authors answered my questions and corrected this manuscript accordingly. After careful consideration, I have no comments to add.

Comments on the Quality of English Language

Line 13: "electroencephalography (EEG) network connections" Electroencephalography is the method of signal acquisition. 

Line 15: "The longer operation time of DE group indicated DE interference spatial cognition resulting in poor performance during spatial cognition task." Please revise for clarity

This manuscript is a resubmission of an earlier submission. The following is a list of the peer review reports and author responses from that submission.

Round 1

Reviewer 1 Report

Comments and Suggestions for Authors

The report by Kai Yang et al. presented the results of an EEG study of humans involved in an operant cognitive task. The authors stated in Abstract that their "results provide a theoretical basis for finding biomarkers of early depression". The major problem is that this study did not focus on depression per se. The authors used a well-known CES-D and BDI questionnaires to measure severity of depressive symptoms. Therefore, their concept of "depressive emotion" looks highly ambiguous and unclear.

CES-D scores in subjects with depressive emotions were 24.8 ±4.2 and in the control group - 14.6 ±3.4. BDI scores in subjects with depressive emotions were 10.9 ±2.6, and in control subjects - 8.2 ±2.1. Results of these questionnaires should be significantly different in different groups, and statistical analysis is required (such as cluster analysis or other classifications).

If the authors would like to refer to a subthreshold depression, they should note that there is no clear consensus on its definition, diagnostic tools, causes, course, outcomes, or management. The criteria used for subthreshold depression are highly heterogeneous, which leads to very heterogeneous epidemiological data [Volz et al. 2022 https://doi.org/10.1080/13651501.2022.2087530]. 

The introduction is too long and it focuses on irrelevant issues, such as depression as a global mental disorder and EEG-related changes in depressed patients.

The authors employed a simulated robotic arm docking task, but they did not explain this task. As it follows from Figures 1 and 2, this task involves a virtual mobile robot arm moving towards and docking with an object or environment.

Lines 37-39. "So, there has been limited research exploring the impact of depressive emotion on cognition, and it remains unclear whether depressive emotion can also interfere with people’s cognitive abilities." 

The authors should do a better literature search before writing this statement.

The results presented in Table 1 are not convincing. The group HC (10 subjects) was compred with the group DE (10 subjects). Did these data pass the normality test? The sample size was not big enough for a parametric t-test, which means the results may not be statistically significant. I am not convinced by the significance of difference between the two groups in reaction time on Stage 2 : 20.32(3.2) and 23.79(2.9).

Discussion describes results, which are missig in Results section.

Based on the said above, I am skeptical about the general conclusion: "results not only expand the understanding of the  brain mechanism of DE affecting cognition, but also are meaningful for finding biomarkers for early diagnosis and intervention of depression."

Comments on the Quality of English Language

The text should be shortened and clarified for better understanding.

In the research of exploring the interaction between emotion and cognition

Some sentences were too simplified and sounded ignorant.

"The brain is a complex system, and the completion of advanced cognitive function requires the interaction and collaborative cooperation of multiple brain regions"

"Therefore, it is difficult to fully analyze the impact of depression on cognition from the perspective of independent brain regions alone."

"Because the rhythm of EEG in different frequency bands contains rich information,"

Author Response

Thank you for your professional opinion! Please see the attachment.

Reviewer 2 Report

Comments and Suggestions for Authors

The authors administered the CES-D and BDI to 20 undergraduate and graduate students, and, in accord with self-scoring, divided them post hoc into two groups of similar size, the depressive emotion (DE) group, which included 10 subjects (4 females), and the healthy control group.  They found that, compared to the healthy control group, the depressive emotion group showed a stronger phase synchronization of the left frontal-parietal network connection. They concluded that an enhanced level of frontal-parietal connections indicates an enhanced executive function, which may compensate for the brain’s spatial cognition deficits and ensure the completion of cognitive tasks, and that their results provide a theoretical basis for finding biomarkers of early depression.

General: the post hoc division into the groups with and without diagnosis has no sense.

Table 1 and figures 3 and 4. Lacking t-values.

Table 2. In Methods, the authors had written “To avoid the deviation caused by multiple comparisons, we utilized false discovery rate (FDR) [57] verification to correct the results of the t-test statistical analysis.” Why don’t they account for number of correlation coefficients in this Table 2? Therefore, it is not true that “the underlined indicates that there is a significant correlation (P<0.05)” for at least two of three underlined coefficients.  And what they want to say by highlighting even lover correlation coefficients (“greater than 0.2”)?

Table 3. A tricky result, nothing more.

Author Response

Thank you for your kind opinion. Please see the attachment.

Reviewer 3 Report

Comments and Suggestions for Authors

The authors present the article entitled “EEG Network Analysis of Depressive Emotion Interference  Cognition Based on a Simulated Robotic Arm Docking Task”

This paper explores the brain mechanism of depressive emotion interference, spatial cognition, and executive function from the brain network perspective. 

The article presents the following general concerns:

  • Improving the keywords and validating the numbers are necessary.

  • Avoid using We instead use the passive voice.

  • Improve the summary by highlighting the most relevant quantitative results.

  • Avoid using contractions.

  • Add hyperlinks to references, tables, and images.

  • Please complement the introduction with previous work and quantitative research results.

  • The introduction needs to be improved since it is not clear what the scientific or technological controversy or contribution is compared to works that attack or attempt to solve the problem raised.

  • At the end of the introduction, briefly describe how the article is structured.

  • Add a brief introduction between titles and subtitles.

  • How representative is the sample selected for the analysis since there are only ten volunteers for both groups? The sample needs to be increased or justified.

  • Lines 215-279 could be justified with the following works: Impact of eeg parameters detecting dementia diseases: a systematic review; A comparative study of time and frequency features for eeg classification; cortical activity at baseline and during light stimulation in patients with strabismus and amblyopia; A new approach for motor imagery classification based on sorted blind source separation, continuous wavelet transform, and convolutional neural network.

  • In addition to the scores of the tests above, what were the inclusion and exclusion criteria of the participants?

  • The paragraph between lines 148 and 177 must be more significant to segment.

  • It is necessary to mention what hardware was used for the tasks.

  • Please explain what is observed in Figures 1 and 2.

  • Please clarify if the EEG signals were obtained during the performance of the cognitive tasks.

  • Justify the use of the Chebyshev filter and how they were dealt with: Ripple in the passband and phase distortion. Also, indicate the order of the filter.

  • Specify how the information was divided for SVM training and validation and whether the number of tests is sufficient for reliable results.

  • Improve the quality of Figures 3 and 4.

  • Avoid repeating the exact Figure and table more than once per paragraph.

  • Please update the references since less than 50% are older than 2018.

  • In the discussions section, add a comparative table between the most representative results of this work and previous works.

  • You have written the same word “offline” with and without a hyphen in your document.Both are acceptable, but it’s best to be consistent. The same case with “preprocess”.

Author Response

(The authors gave the same response as above.)
